# Post-Transplant Cyclophosphamide Combined with Brilliant Blue G Reduces Graft-versus-Host Disease without Compromising Graft-versus-Leukaemia Immunity in Humanised Mice

**DOI:** 10.3390/ijms25031775

**Published:** 2024-02-01

**Authors:** Peter Cuthbertson, Amy Button, Chloe Sligar, Amal Elhage, Kara L. Vine, Debbie Watson, Ronald Sluyter

**Affiliations:** 1Molecular Horizons and School of Chemistry and Molecular Bioscience, University of Wollongong, Wollongong, NSW 2522, Australia; cpeter@uow.edu.au (P.C.); ab166@uowmail.edu.au (A.B.); cs821@uowmail.edu.au (C.S.); ae880@uowmail.edu.au (A.E.); kara@uow.edu.au (K.L.V.); dwatson@uow.edu.au (D.W.); 2Illawarra Health and Medical Research Institute, Wollongong, NSW 2522, Australia

**Keywords:** purinergic receptor, P2X7, post-transplant cyclophosphamide, xenogeneic graft-versus-host disease, graft-versus-tumour immunity, acute myeloid leukaemia, lymphocyte, ectonucleotidase, cytokine, adverse event

## Abstract

Allogeneic haematopoietic stem cell transplantation (HSCT) leads to the establishment of graft-versus-leukaemia (GVL) immunity, but in many cases also results in the development of graft-versus-host disease (GVHD). This study aimed to determine if P2X7 antagonism using Brilliant Blue G (BBG) could improve the beneficial effects of post-transplant cyclophosphamide (PTCy) in a humanised mouse model of GVHD, without comprising GVL immunity. NOD.Cg-*Prkdc^scid^ Il2rg^tm1Wjl^* (NSG) mice were injected with human peripheral blood mononuclear cells (PBMCs) (Day 0), then with cyclophosphamide (33 mg/kg) on Days 3 and 4, and with BBG (50 mg/kg) (or saline) on Days 0–10. PTCy with BBG reduced clinical GVHD development like that of PTCy alone. However, histological analysis revealed that the combined treatment reduced liver GVHD to a greater extent than PTCy alone. Flow cytometric analyses revealed that this reduction in liver GVHD by PTCy with BBG corresponded to an increase in human splenic CD39^+^ Tregs and a decrease in human serum interferon-γ concentrations. In additional experiments, humanised NSG mice, following combined treatment, were injected with human THP-1 acute myeloid leukaemia cells on Day 14. Flow cytometric analyses of liver CD33^+^ THP-1 cells showed that PTCy with BBG did not mitigate GVL immunity. In summary, PTCy combined with BBG can reduce GVHD without compromising GVL immunity. Future studies investigating P2X7 antagonism in combination with PTCy may lead to the development of novel treatments that more effectively reduce GVHD in allogeneic HSCT patients without promoting leukaemia relapse.

## 1. Introduction

Allogeneic haematopoietic stem cell transplantation (alloHSCT) is a life-saving therapy for people with blood cancers, often used for the treatment of acute myeloid leukaemia (AML) [1]. The effectiveness of alloHSCT in treating blood cancers is due to the establishment of graft-versus-leukaemia (GVL) immunity, in which donor CD4^+^ T cells and CD8^+^ T cells, as well as donor natural killer (NK) cells, eliminate host malignant cells [2]. However, the efficacy of alloHSCT can be mitigated by the development of acute and chronic graft-versus-host disease (GVHD) [3].

Acute GVHD (hereafter termed GVHD) is mediated by the presentation of alloantigens by host and donor dendritic cells to donor CD4^+^ T cells, including T helper (Th) 1 cells and Th17 cells, and CD8^+^ T cells, which can damage the liver, lung, gut, skin, and other organs of alloHSCT recipients [4]. The pro-inflammatory and cytotoxic action of donor T cells in mediating GVHD can be prevented in part by the presence of regulatory T cells (Tregs) [4]. Clinically, GVHD can be prevented by different forms of prophylaxis, with post-transplant cyclophosphamide (PTCy) commonly used in haploidentical (half-matched) alloHSCT recipients, with emerging use in matched related or unrelated or mismatched unrelated alloHSCT recipients [5]. Despite the use of PTCy and other forms of prophylaxis, GVHD still occurs in around 30% of blood cancer patients following alloHSCT [3]. Thus, there is an urgent need to identify new drug targets and develop more effective treatments to prevent GVHD without compromising GVL immunity.

The P2X7 receptor has emerging roles in the development of GVHD [6,7], with various inhibitors of P2X7 able to reduce GVHD in different mouse models of this disease [8,9,10,11,12]. P2X7 is a trimeric ligand-gated ion channel which is present on immune cells and activated by extracellular adenosine 5′-triphophate (ATP) to promote inflammation and immunity [13]. P2X7 activation stimulates various functions relevant to the development of GVHD. For example, P2X7 activation contributes to the function of dendritic cells, including cross-dressing and presentation of antigens [14], the up-regulation of co-stimulatory molecules [15], and the release of pro-inflammatory cytokines [16]. P2X7 activation also contributes to the differentiation of Th1 cells [17] and Th17 cells [18]. Further to this, P2X7 activation can inhibit the generation and function of Tregs, and promote their conversion to Th17 cells [19].

Humanised mouse models of GVHD, in which human peripheral blood mononuclear cells (PBMCs) are injected into immunodeficient mice, most commonly NOD.Cg-*Prkdc^scid^ Il2rg^tm1Wjl^* (NSG) mice, are a valuable pre-clinical model to test and explore the efficacy of existing and emerging GVHD therapies [20]. To this end, our group has demonstrated that cyclophosphamide (33 mg/kg) administered intraperitoneally (i.p.) on Days 3 and 4 post-human PBMC injection reduces but does not prevent the development of GVHD in humanised NSG mice [21,22]. Moreover, our group has shown that the P2X7 antagonist Brilliant Blue G (BBG) (50 mg/kg) administered i.p. daily on Days 0–10 post-human PBMC injection can also reduce GVHD development in these mice [23]. However, the effectiveness of these two prophylactic treatments in combination has not been investigated. Therefore, the current study aimed to determine if P2X7 antagonism using BBG could improve the beneficial effects of PTCy without comprising GVL immunity.

## 2. Results

### 2.1. PTCy with BBG Does Not Reduce Clinical GVHD Compared to PTCy Alone

Our group has shown previously that either PTCy [21,22] or the P2X7 antagonist BBG [23] reduces GVHD in humanised NSG mice. Therefore, to determine if PTCy with P2X7 antagonism could further reduce GVHD, humanised NSG mice were injected i.p. with cyclophosphamide and BBG (PTCy + BBG mice) or cyclophosphamide and saline (BBG vehicle) (PTCy mice) at the same doses and timings as used in these former studies. Mice were monitored three times per week until humane (disease) endpoint or to Day 70, the experimental endpoint (Figure 1a). Both groups of mice gained weight until Day 30, with weight values remaining constant until experimental endpoint and with no significant difference between the groups (*p* = 0.37) (Figure 1b). From Day 20, clinical scores increased in both groups, with scores continuing to increase in PTCy mice but not in PTCy + BBG mice, which plateaued from Day 50 (Figure 1c). There was no significant difference in clinical scores over time between the two groups (*p* = 0.48). Ear thickness, used to assess skin GVHD [24], was similar between groups (*p* = 0.70) (Figure 1d). Overall survival was also similar between PTCy and PTCy + BBG mice, both with median survival times of >70 days (*p* = 0.30). However, the percentage of survival on Day 70 for PTCy + BBG mice (80%) was about twice that of PTCy mice (54%), a difference that approached statistical significance (*p* = 0.10) (Figure 1e).

### 2.2. PTCy with BBG Reduces Liver GVHD Compared to PTCy Alone at Endpoint

We have shown previously that PTCy reduces liver GVHD in humanised NSG mice [21], and that BBG reduces GVHD in both the liver and skin of these mice [23]. Therefore, the effect of PTCy with BBG compared to PTCy on histological GVHD was assessed at endpoint. PTCy mice displayed mild to severe histological GVHD with immune cell infiltration observed in the livers, flank skin, ears, and lungs (Figure 2). In addition, some mice displayed increased epidermal thickening and apoptotic bodies in the skin and ears. Similar results were observed in these organs from PTCy + BBG mice, except for the liver, where the histological grade was reduced by ~65% in PTCy + BBG mice compared to PTCy mice (*p* < 0.05). Lung GVHD was quantified as percent clear alveoli space and was found to be similar between both groups (*p* = 0.95). In contrast to these organs, histological GVHD in the duodenum was minimal, with the histological grade alike between both groups (Figure 2).

### 2.3. PTCy with BBG Reduces Serum Human Interferon-γ Concentrations Compared to PTCy Alone at Endpoint

BBG increases proportions of human Tregs and decreases human serum interferon-γ (IFNγ) concentrations in humanised NSG mice [23]. Therefore, the effect of PTCy with BBG compared to PTCy alone on human splenic immune cells and serum IFNγ and 12 other pro-inflammatory and regulatory human cytokines were assessed at endpoint by flow cytometry. The flow cytometry gating strategy for cells has been outlined previously [23]. The proportions of human CD45^+^ leukocytes (*p* = 0.51) (Figure 3a), human CD3^+^ T cells (*p* = 0.50) (Figure 3b), and human CD4^+^ or CD8^+^ T cells (*p* > 0.99) (Figure 3c) were similar between both groups. The human CD4^+^:CD8^+^ T cell ratios were also comparable between both groups (*p* = 0.35) (Figure 3d). The proportions of human Tregs were similar between both groups (*p* = 0.51) (Figure 3e). Likewise, proportions of human CD39^+^ Tregs, which mediate immunosuppression via hydrolysis of extracellular ATP [25,26], were similar between both groups (*p* = 0.92) (Figure 3f). Proportions of human CD19^+^ B cells, which were relatively minor (<0.5%), were not significantly increased in PTCy + BBG mice compared to PTCy mice (*p* = 0.11) (Figure 3g).

A flow cytometric multiplex analysis of sera collected at endpoint revealed that human IFNγ concentrations were reduced by 40% in PTCy + BBG mice compared to PTCy mice (*p* < 0.01) (Figure 3h). In contrast, concentrations of the human cytokines interleukin (IL)-2 (*p* = 0.96), IL-4 (*p* = 0.80), IL-5 (*p* = 0.20), IL-6 (*p* = 0.40), IL-9 (*p* = 0.46), IL-10 (*p* = 0.21), IL-13 (*p* = 0.19), IL-17A (*p* = 0.44), IL-17F (*p* = 0.74), IL-21 (*p* = 0.98), IL-22 (*p* = 0.13), and tumour necrosis factor-α (TNFα) (*p* = 0.72) were similar between both groups (Appendix A).

### 2.4. PTCy with BBG Increases Human CD39^+^ Tregs Compared to PTCy Alone on Day 21

Our previous studies have revealed immunological changes when studying humanised NSG mice on Day 21 [8,23]. To compare the effect of PTCy with BBG to PTCy alone on early GVHD development in humanised NSG mice, the above regimens (Section 2.1) were used but with mice sacrificed on Day 21 (Figure 4a). Organ samples were assessed by histology and flow cytometry as described above (Section 2.2 and Section 2.3). As previously observed [8,23], histological GVHD in the liver, skin, ear, lung, and duodenum from both groups was minimal at this timepoint (Appendix A).

Flow cytometric analyses of spleen samples on Day 21 revealed that the proportions of human CD45^+^ leukocytes (*p* = 0.89) (Figure 4b), CD3^+^ T cells (*p* = 0.39) (Figure 4c), and CD4^+^ and CD8^+^ T cells (*p* > 0.99) (Figure 4d) were similar between both groups. Moreover, the human CD4^+^:CD8^+^ T cell ratios were comparable between both groups (*p* = 0.93) (Figure 4e). In contrast, proportions of human Tregs were increased approximately two-fold in PTCy + BBG mice (*p* = 0.065), with 40% of these mice having at least two-fold greater proportions of these cells compared to those in PTCy mice (Figure 4f). Furthermore, the proportions of human CD39^+^ Tregs were increased more than two-fold in PTCy + BBG mice compared to PTCy mice (*p* < 0.001) (Figure 4g). Proportions of human CD19^+^ B cells were comparable between both groups on Day 21 (*p* = 0.48) (Figure 4h) but greater than that observed on Day 70 above (Figure 3g). Despite not being detected in spleens on Day 70, human NK cells were present on Day 21 and proportions were similar between both groups (*p* = 0.48) (Figure 4i).

Finally, flow cytometric analyses of sera samples on Day 21 revealed the concentrations of human IFNγ were not significantly different between both groups (*p* = 0.63), with most samples below the detection limit of the assay (Figure 4h). Furthermore, the concentrations of human IL-2 (*p* = 0.48), IL-4 (*p* = 0.14), IL-5 (*p* > 0.99), IL-6 (*p* = 0.96), IL-9 (*p* = 0.85), IL-10 (*p* not assessable), IL-13 (*p* = 0.47), IL-17A (*p* = 0.74), IL-17F (*p* not assessable), IL-21 (*p* > 0.99), IL-22 (*p* = 0.59) and TNFα (*p* = 0.09) from both groups were similar, with these cytokines below the detection limit of the assay in most samples (Appendix A).

### 2.5. PTCy with BBG Does Not Compromise GVL Immunity

A key goal of alloHSCT in people with blood cancers is the establishment of GVL immunity and to ensure any GVHD prophylaxis does not compromise this effect [2]. PTCy has been shown recently not to compromise GVL immunity in humanised NSG mice [27], but the impact of P2X7 antagonism on this effect in this mouse model has not been reported. To compare the impact of PTCy with BBG on GVL immunity, NSG mice were injected i.p. with human PBMCs or Dulbecco’s phosphate-buffered saline (PBS) on Day 0. These mice, with or without PBMCs, were then injected i.p. with cyclophosphamide on Days 3 and 4 and BBG on Days 0–10 as described above (Section 2.1 and Section 2.4) or with equal volumes of the corresponding diluents, PBS and saline, respectively. All mice were then injected intravenously (i.v.) with 1 × 10^6^ human AML THP-1 cells on Day 14 and monitored three times per week until humane (disease) endpoint or to Day 42, the experimental endpoint of this GVHD/GVL model (Figure 5a). Mice injected with both PBMCs and THP-1 cells and treated with PTCy and BBG or PBS and saline are described as GVL + PTCy + BBG and GVL mice, respectively. Mice injected with THP-1 cells but not PBMCs and treated with PTCy and BBG or PBS and saline were designated THP-1 + PTCy + BBG mice and THP-1 mice, respectively.

All groups of mice gained weight until Day 17, with GVL + PTCy + BBG and GVL mice progressively losing ~10% of weight from Days 20 and 17, respectively with no differences in weight between these two groups (*p* = 0.84) (Figure 5b). In contrast, THP-1 + PTCy + BBG and THP-1 mice continued to gain weight in a similar manner until endpoint (*p* = 0.99). Clinical scores in GVL + PTCy + BBG and GVL mice progressively increased over time, with the average clinical score lower in GVL + PTCy + BBG mice compared to GVL mice (*p* = 0.09) (Figure 5c). In contrast, THP-1 + PTCy + BBG and THP-1 mice had minimal and similar clinical scores over time (*p* = 0.44). Clinical scores paralleled median survival times of >42 days for GVL + PTCy + BBG mice and 34.5 days for GVL mice (*p* = 0.25), and median survival times of >42 days for both THP-1 + PTCy + BBG and THP-1 mice (Figure 5d). The percentage of survival on Day 42 was two-fold that in GVL + PTCy + BBG mice (56%) compared to GVL mice (25%) (*p* = 0.20). All THP-1 + PTCy + BBG and THP-1 mice survived until Day 42.

One mouse from each of the GVL, THP-1 + PTCy + BBG, and THP-1 groups developed a curled hind paw during the study. This adverse event subsequently resolved. The GVL mouse however was found deceased on Day 19 and was excluded from the above and subsequent analyses. Spleens and faeces collected from the other two mice at endpoint were examined for lactate dehydrogenase elevating virus, associated with myelopathy and neuropathy in NSG mice [28], with both mice testing negative for this virus, and thus included in all analyses.

The liver is a major site of human THP-1 AML cell accumulation in NSG mice [29]. Flow cytometric analyses of this organ at endpoint revealed that human CD33^+^ THP-1 cells were present and at similar proportions in THP-1 + PTCy + BBG and THP-1 mice (*p* = 0.53) but were minimal in GVL + PTCy + BBG and GVL mice (*p* > 0.99) (Figure 5e). Notably, proportions of human CD33^+^ THP-1 cells were significantly lower in GVL + PTCy + BBG and GVL mice versus THP-1 + PTCy + BBG (*p* = 0.01) and THP-1 mice (*p* = 0.0002), respectively.

### 2.6. PTCy with BBG Does Not Alter Proportions of Human Immune Cell Subsets Compared to PTCy in GVL Immunity at Endpoint

Flow cytometric analyses of spleens from mice injected with PBMCs at endpoint revealed the proportions of human CD45^+^ cells (comprising human leukocytes and possibly human THP-1 cells) were significantly lower in GVL + PTCy + BBG mice compared to GVL mice (*p* < 0.05) (Figure 6a). In contrast, proportions of human CD3^+^ T cells (*p* = 0.48) (Figure 6b) and human CD4^+^ or CD8^+^ T cells (*p* = 0.84 and *p* = 0.75, respectively) (Figure 6c) were similar between both groups. The human CD4^+^:CD8^+^ T cell ratios were also similar between both groups (*p* = 0.54) (Figure 6d). The proportions of human Tregs were increased 1.4-fold in GVL + PTCy + BBG mice compared to GVL mice, a difference that approached statistical significance (*p* = 0.065) (Figure 6e). In contrast, proportions of human CD39^+^ Tregs were similar between both groups (*p* = 0.75) (Figure 6f).

Flow cytometric analyses of livers from GVL + PTCy + BBG and GVL mice at endpoint revealed similar proportions of human CD45^+^ cells (*p* = 0.28) (Figure 7a), human CD3^+^ T cells (*p* = 0.75) (Figure 7b), and human CD4^+^ or CD8^+^ T cells (*p* = 0.69 and *p* > 0.99, respectively) (Figure 7c). The human CD4^+^:CD8^+^ T cell ratios were also similar between both groups (*p* = 0.96) (Figure 7d), as were the proportions of human Tregs (*p* = 0.68) (Figure 7e) and human CD39^+^ Tregs (*p* = 0.41) (Figure 7f).

## 3. Discussion

This study demonstrates that PTCy combined with the P2X7 antagonist BBG reduced clinical GVHD development to an extent like that of PTCy alone in humanised NSG mice. Notably, the combined treatment reduced liver GVHD to a greater extent than PTCy alone. This reduction in liver GVHD by PTCy with BBG corresponded to an increase in human splenic CD39^+^ Tregs on Day 21 and a decrease in human serum IFNγ concentrations at endpoint. Importantly, PTCy with BBG did not compromise GVL immunity to human THP-1 AML cells in humanised NSG mice, despite partly reducing proportions of total human leukocytes.

PTCy alone impaired clinical GVHD in humanised NSG mice with a comparable efficacy to that observed previously in this model [21,22]. Although a control group, namely humanised NSG mice injected with PBS on Days 3 and 4, was not included to reduce mouse numbers as per the 3Rs [30], weight gain, clinical scores, and survival followed similar patterns to those seen previously in humanised NSG mice injected with PTCy (33 mg/kg) i.p. on Days 3 and 4 [21,22]. These current and past observations also parallel recent observations in which humanised NSG mice injected i.p. with PTCy (100 mg/kg) on Day 3 also display reduced clinical GVHD [27]. Collectively, these findings support the use of this preclinical mouse model to study the mechanism of action of PTCy, as well as the study of this prophylactic treatment in combination with other GVHD treatments, as previously shown for PTCy with Tocilizumab [22].

PTCy with BBG did not reduce clinical GVHD development any further than that of PTCy alone in humanised mice. This may be because of the PTCy regimen used to limit GVHD progression having near maximal efficacy in this model. PTCy at 75 mg/kg on Days 3 and 4, but not at 25 mg/kg on Days 3 and 4, reduces GVHD progression in humanised NSG mice [31]. This suggests a dose-dependent effect of this treatment in preventing GVHD, consistent with an allogeneic mouse model of GVHD [32]. The limited impact of PTCy with BBG compared to PTCy alone may also relate to the mechanism of actions of these treatments. Both drugs may potentially target similar stages of GVHD development, namely the afferent phase, in which dendritic cells are activated, and the first part of the efferent phase, in which T cells are activated and subsequently proliferate [33]. The precise mechanism of action of either PTCy or P2X7 antagonism remains to be fully defined. To date, PTCy has been shown to deplete proliferating human CD3^+^ T cells [21] and more specifically human CD4^+^ and CD8^+^ T cells [27]. PTCy may also deplete human Tregs [21,27] but this has not been observed in all studies [22]. In regards to P2X7, activation of this receptor is involved in the stimulation of host dendritic cells in an allogeneic mouse model [12], while studies of both allogeneic and humanised mouse models support a role for donor P2X7 activation in the loss of donor Tregs [8,12,23]. Moreover, allogeneic mouse models have revealed roles for both PTCy and P2X7 activation in the expansion [34] and dysfunction [11] of myeloid-derived suppressor cells, respectively, which can alter Tregs [34]. Regardless of the reasons for the lack of improvement in clinical GVHD by PTCy with BBG compared to PTCy alone, this study reveals that P2X7 antagonism does not mitigate the efficacy of PTCy, keeping open the possibility that this combined treatment may be of value to prevent GVHD, especially liver GVHD, in patients following alloHSCT.

PTCy with BBG reduced liver GVHD in humanised NSG mice compared to PTCy alone. This supports several observations in which P2X7 blockade, by either BBG or an anti-human P2X7 monoclonal antibody (mAb), reduces liver GVHD in allogeneic [35] and humanised [8,10,23,36] mouse models. Why the liver, compared to other GVHD organs, repeatedly benefits from P2X7 blockade during GVHD progression is not known. However, the reduced liver GVHD in these previous studies often corresponded to increased proportions of Tregs and reduced serum IFNγ concentrations, suggesting that these two aspects of the immune system may be linked to reduce liver GVHD. Consistent with this, PTCy with BBG increased proportions of human Tregs, most notably CD39^+^ Tregs, and reduced human serum IFNγ concentrations compared to PTCy alone. Whether these two observations are directly related remains unknown but lack of IFNγ receptor signalling has been associated with increased Tregs and reduced GVHD in allogeneic and humanised mice [37]. This suggests that reduced human IFNγ concentrations may facilitate the increase in human Tregs in the current study. However, it should be noted that concentrations of IFNγ or other human cytokines in the liver were not determined. A further explanation as to why the liver repeatedly benefits from P2X7 blockade may be due to BBG or the anti-human P2X7 mAb accumulating in the liver at higher concentrations than elsewhere. However, if true, this does not explain the limited effectiveness of PTCy in reducing liver GVHD [22], which is largely metabolised to its active metabolites within the liver [38,39]. Regardless, P2X7 antagonism may benefit patients following alloHSCT by curtailing liver GVHD. This form of GVHD occurs in 16% of patients with post-alloHSCT liver dysfunction and is associated with increased non-relapse mortality and poor survival [40]. Thus, such patients would benefit from new treatments to prevent or reduce this form of GVHD.

PTCy with BBG did not compromise GVL immunity to human THP-1 AML cells in humanised NSG mice. Consistent with this, PTCy in humanised NSG mice [27] or P2X7 antagonism with pyridoxalphosphate-6-azophenyl-2′,4′-disulfonic acid [12] in allogenic mice reduces GVHD without comprising GVL immunity to human THP-1 AML or murine A20 lymphoma cells, respectively. Moreover, PTCy with BBG did not alter proportions of human immune cell subsets in humanised NSG mice compared to those mice without PTCy and BBG. Notably, this was observed for human CD4^+^ and CD8^+^ T cells, effector cells important in mediating GVL immunity [2]. Proportions of human Tregs were partly increased in mice treated with PTCy and BBG compared to those without PTCy and BBG, but based on the collective data of this study this appears to impair GVHD rather than have any major impact on GVL immunity.

Although PTCy with BBG did not compromise GVL immunity or alter human immune cell subset proportions in humanised mice, this combined treatment reduced the proportions of total human leukocytes compared to control treated humanised mice. Our previous studies revealed that PTCy alone does not alter proportions of these cells [21,22], while others have shown that PTCy reduces proportions of human leukocytes at early but not later timepoints in humanised mice [27,31]. Thus, these differences likely reflect the timepoint at which human leukocytes were examined. Likewise, BBG can reduce proportions of human leukocytes at an earlier but not later timepoint [23]. In contrast, a blocking anti-human P2X7 mAb does not alter proportions of these cells at early or late timepoints [8]. This suggests that P2X7 antagonism with BBG, which can inhibit both human and mouse P2X7 [41], may impair the engraftment of human leukocytes, shortly after human PBMC injection, by inhibiting P2X7 on mouse cells, which regulate engraftment of these cells in an unknown way. Additionally, the reduced proportions of human leukocytes may be due to reduced proliferation of xenoreactive human T cells, because of BBG inhibiting P2X7 on mouse dendritic cells and subsequently limiting human T cell activation.

Finally, in NSG mice that did not receive human PBMCs, PTCy with BBG did not reduce proportions of human CD33^+^ THP-1 cells compared to those without PTCy and BBG. This indicates that the combined treatment does not directly impact THP-1 leukaemia development. These findings are consistent with the relatively fast (within hours) clearance of cyclophosphamide and its metabolites from mice [42,43], which suggest these compounds are not present when THP-1 cells are injected. The lack of effect of BBG on THP-1 cells is consistent with the absence of functional P2X7 in these cells [44,45], suggesting that this drug cannot directly affect these cells by blocking P2X7. P2X7 is reported to be present on some primary human AML cells [46,47], but it remains unknown if the use of P2X7 antagonism in patients with blood cancers to prevent GVHD would increase cancer relapse. However, P2X7 knockdown [46] or antagonism [47] reduces leukaemia growth in mice, so the use of a P2X7 antagonist or inhibitory mAb may have a dual role in both preventing GVHD whilst reducing leukaemia in blood cancer patients following alloHSCT. A limitation of our study is that the THP-1 cell model does not fully recapitulate human AML, with the liver the main site of THP-1 cell accumulation in NSG mice [29].

In conclusion, PTCy combined with BBG reduces GVHD progression without compromising GVL immunity to human THP-1 AML cells. Moreover, this combined treatment reduces liver GVHD compared to PTCy alone, an effect that corresponds with an increase in human CD39^+^ Tregs and a decrease in human serum IFNγ. Future studies investigating P2X7 antagonism in combination with PTCy may lead to the development of novel treatments that more effectively reduce GVHD in allogeneic HSCT patients without promoting leukaemia relapse.

## 4. Materials and Methods

### 4.1. Materials

VACUETTE lithium heparin tubes were obtained from Greiner Bio-One (Frickenhausen, Germany). Ficoll-Paque PLUS was obtained from GE Healthcare (Uppsala, Sweden). PBS, RPMI-1640 medium, GlutaMAX, and FCS (heat-inactivated at 56 °C for 30 min) were obtained from Thermo Fisher Scientific (Waltham, MA, USA). Sterile isotonic saline was obtained from Pfizer (Sydney, NSW, Australia). BBG, cyclophosphamide, and 10% neutral-buffered formalin were obtained from Sigma-Aldrich (St Louis, MO, USA). Sterile BBG (5 mg/mL) was prepared in isotonic saline, and sterile cyclophosphamide (6.6 mg/mL) was prepared in PBS. Haematoxylin and eosin stain was obtained from POCD Scientific (Artarmon, NSW, Australia). Zombie NIR live/dead stain was obtained from BioLegend (San Diego, CA, USA). Fluorochrome-conjugated mAbs, as described in detail [24], and R-phycoerythin-conjugated anti-human CD33 mAb (clone P67.6) were obtained from BD Biosciences (San Diego, CA, USA).

### 4.2. Human Cells

Human peripheral blood and PBMCs were obtained and used as approved by the Human Ethics Research Committee, University of Wollongong (Wollongong, NSW, Australia). Peripheral blood from healthy donors (aged 23–27 years; three males and one female) was collected into VACUETTE lithium heparin tubes. Human PBMCs were isolated by Ficoll-Paque density centrifugation as described [48].

THP-1 cells (American Type Culture Collection, Manassas, VA, USA) were cultured in RPMI-1640 medium containing 2 mM GlutaMAX and 10% FCS at 37 °C and 5% CO_2_/95% air. THP-1 cells were 100% identical to reference THP-1 cells, determined by short tandem repeat analysis (Garvan Molecular Genetics, Sydney, NSW, Australia). THP-1 cells were negative for mycoplasma as determined by the MycoAlert Test Kit (Lonza, Basel, Switzerland).

### 4.3. NSG Mice

NSG mice were obtained and used as approved by the Animal Ethics Committee, University of Wollongong. Female NSG mice (aged 4–6 weeks) were obtained from Australian BioResources (Moss Vale, NSW, Australia) or Animal Resources Centre (Canning Vale, WA, Australia). Mice were housed in environmentally enriched ventilated cages (Techniplast, Buggugiate, Italy) with a 12 h light-12 h dark cycle and supplied ad libitum with autoclaved water and food. Mice were acclimatised for at least 2 weeks prior to study.

### 4.4. Humanised Mouse Model of GVHD

NSG mice were injected i.p. with 20 × 10^6^ human PBMCs in PBS (Day 0) and cyclophosphamide (33 mg/kg) or an equivalent volume of PBS (Days 3 and 4). These NSG mice were also injected i.p. daily with BBG (50 mg/kg) or an equivalent volume of saline (Days 0–10). Mice were monitored for GVHD at least three times a week from Day 0 until endpoint (humane endpoint, Day 21 or Day 70) using a GVHD scoring system as described [48]. Ear thickness was measured three times a week using Interapid spring-loaded calipers (Rolle, Switzerland). Weights, clinical scores, and ear thickness measurements were carried forward for euthanised mice until experimental endpoint. Humanised mice were euthanised by slow-fill CO_2_ at endpoint, and organs and blood were collected as indicated. Serum was obtained by centrifugation of blood at 1700× *g* for 10 min and stored at −80 °C until analyses. Faeces and spleens of two mice at endpoint were examined for lactate dehydrogenase elevating virus by polymerase chain reaction (Cerberus Sciences, Adelaide, SA, Australia).

### 4.5. Humanised Mouse Model of GVL Immunity

NSG mice were injected i.p. with 20 × 10^6^ human PBMCs in PBS (Day 0) or PBS. Mice were then injected with cyclophosphamide and BBG as described above (Section 4.4) or equivalent volumes of PBS and saline, respectively. All mice were injected i.v. with 1 × 10^6^ THP-1 cells in PBS (Day 14). Mice were monitored for GVHD at least three times a week from Day 0 until endpoint (humane endpoint or Day 42) using a GVHD/leukaemia scoring system (Table 1). Mice were euthanised and tissues collected as described above (Section 4.4).

### 4.6. Histological Analysis of Tissues

Formalin-fixed tissue sections (3 µm) were stained with haematoxylin and eosin using standard techniques. Sections were assessed in a blinded manner using a Leica (Wetzlar, Germany) DM750 inverted light microscope captured images as described [23].

### 4.7. Flow Cytometric Analysis of Cells

Mechanically isolated spleen and enzymatically isolated liver cell suspensions from organs initially collected into ice-cold PBS were prepared as described [24]. Cells were stained with Zombie NIR live/dead stain, then with fluorochrome-conjugated mAbs, as indicated using standard techniques. Events were acquired using a BD LSR Fortessa X-20 flow cytometer. Proportions of immune and THP-1 cells were assessed using BD FlowJo software version 8.7.1.

### 4.8. Flow Cytometric Analysis of Human Cytokines

Serum human cytokine concentrations were determined using a LEGENDplex Human Th Cytokine Panel (13-plex) kit (BioLegend) according to the manufacturer’s instructions with an Attune NxT Flow Cytometer and Autosampler (Thermo Fisher Scientific). Cytokine concentrations outside the standard curves were set to the corresponding detection limits for statistical analyses.

### 4.9. Data Presentation and Statistical Analysis

Data were analysed using GraphPad Prism (GraphPad Software, Boston, MA, USA, version 8.4.2). Data are reported as mean ± standard error of the mean (SEM) unless stated otherwise. Data sets were tested for normality using the Shapiro–Wilk normality test. Differences between single comparisons were determined using an unpaired Student’s or Welch’s *t* test. Differences between multiple comparisons were determined using a one-way ANOVA (Tukey’s post hoc test) or a Kruskal–Wallis test (Dunn’s post hoc test). Differences between groups for clinical score, weight, and ear thickness over time were determined using a repeated-measures two-way ANOVA with the Geisser–Greenhouse correction. Differences in survival and mortality were determined with a log-rank Mantel–Cox test and Chi-squared (χ^2^) test, respectively. Grubb’s test was used to remove two outliers in the analysis of human IFNγ. *p* < 0.05 was considered statistically significant.

## Figures and Tables

**Figure 1 ijms-25-01775-f001:**
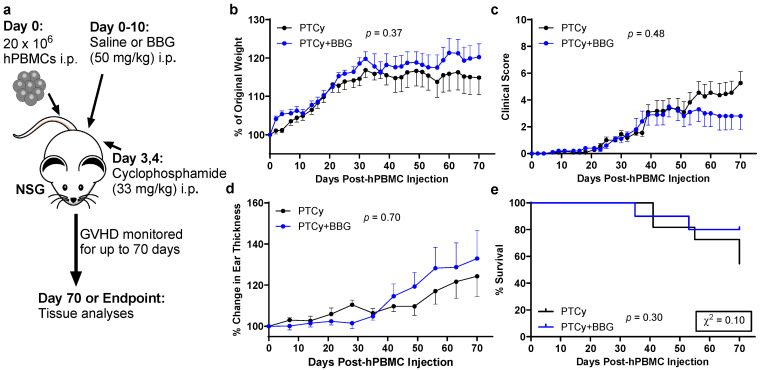
PTCy with BBG does not reduce clinical GVHD compared to PTCy alone. (**a**) Schematic representation of the humanised mouse model of GVHD. (**a**–**e**) NSG mice were humanised with 20 × 10^6^ human PBMCs (hPBMCs) (*n* = 3 donors) on Day 0, treated with cyclophosphamide (33 mg/kg) (Day 3 and 4), and BBG (50 mg/kg) (PTCy + BBG) (*n* = 10 mice) or saline (PTCy) (*n* = 11 mice) (Days 0–10). Mice were assessed thrice weekly for (**b**) weight change, (**c**) clinical score, (**d**) ear thickness, and (**e**) survival. (**b**–**d**) Data are presented as mean ± SEM. (**e**) Data are presented as percent survival, χ^2^ test for mortality.

**Figure 2 ijms-25-01775-f002:**
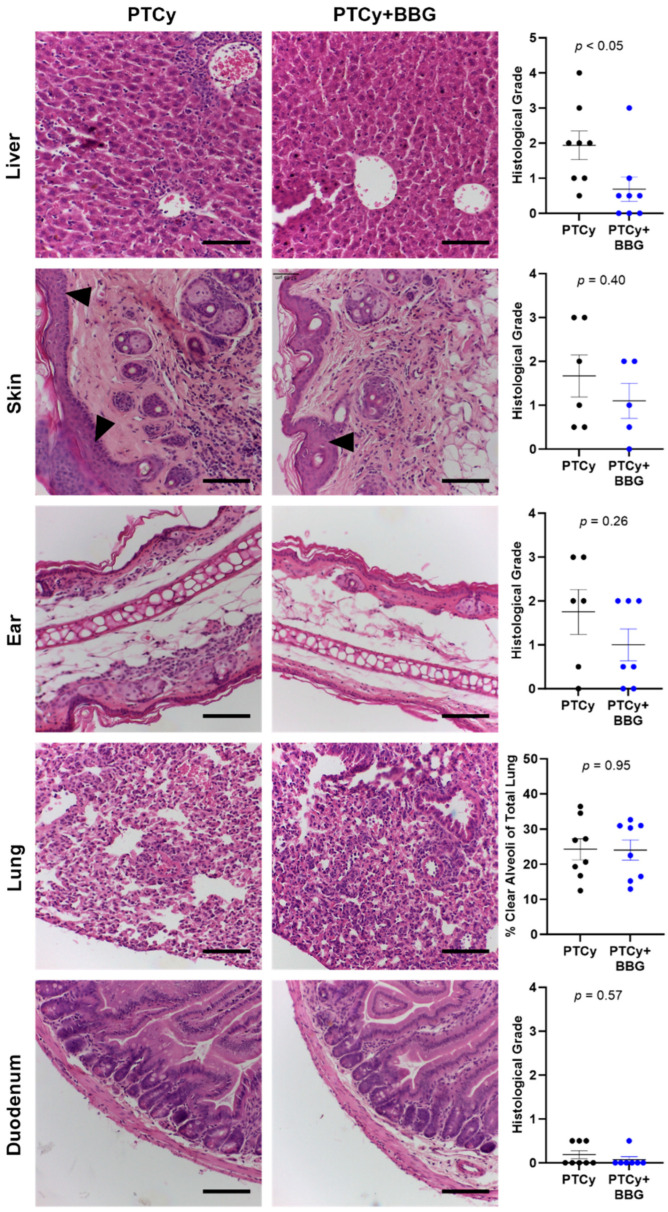
PTCy with BBG reduces liver GVHD compared to PTCy alone at endpoint. Haematoxylin and eosin-stained liver, skin, ear, lung, and duodenum tissue sections from treated humanised mice at endpoint (Figure 1) were examined for evidence of histological GVHD. Histological GVHD in the liver, skin, ear, and duodenum was assessed using a standardised grading system. Histological lung GVHD was determined as the percent of clear alveoli area of total lung area. Arrowheads indicate epidermal thickening in the skin. Images representative of 5–8 mice per treatment group. Scale bars represent 100 µm. Data are presented as mean ± SEM. Symbols represent individual mice.

**Figure 3 ijms-25-01775-f003:**
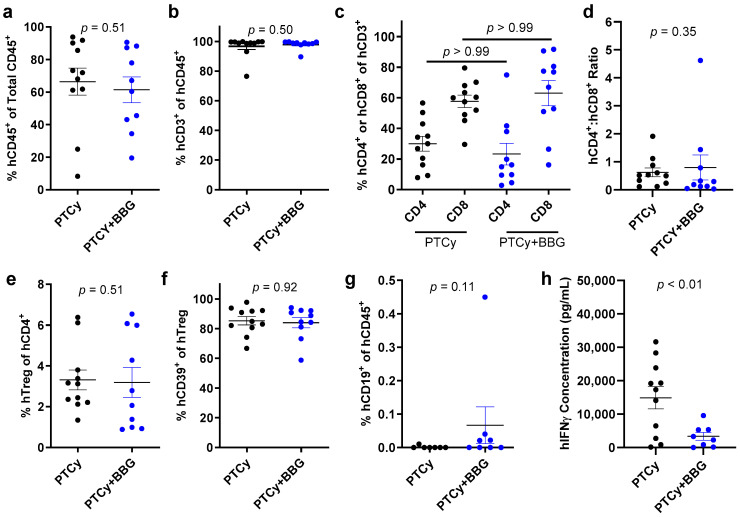
PTCy with BBG reduces serum human IFNγ concentrations compared to PTCy alone at endpoint. (**a**–**h**) Spleen suspensions from treated humanised mice at endpoint (Figure 1) were examined by flow cytometry to determine the proportions of (**a**) human (h) CD45^+^ leukocytes, (**b**) hCD3^+^ T cells, (**c**) hCD4^+^ or hCD8^+^ T cells, (**e**) hCD4^+^hCD25^+^hCD127^lo^ regulatory T cell (hTreg), (**f**) hCD39^+^ Tregs, or (**g**) hCD19^+^ B cells. (**d**) The ratio of hCD4^+^ to hCD8^+^ T cells was calculated from (**c**). (**h**) Serum hIFNγ concentrations were examined by a flow cytometric LEGENDPlex kit. (**a**–**h**) Data are presented as mean ± SEM. Symbols represent individual mice (*n* = 10, PTCy + BBG; *n* = 11, PTCy).

**Figure 4 ijms-25-01775-f004:**
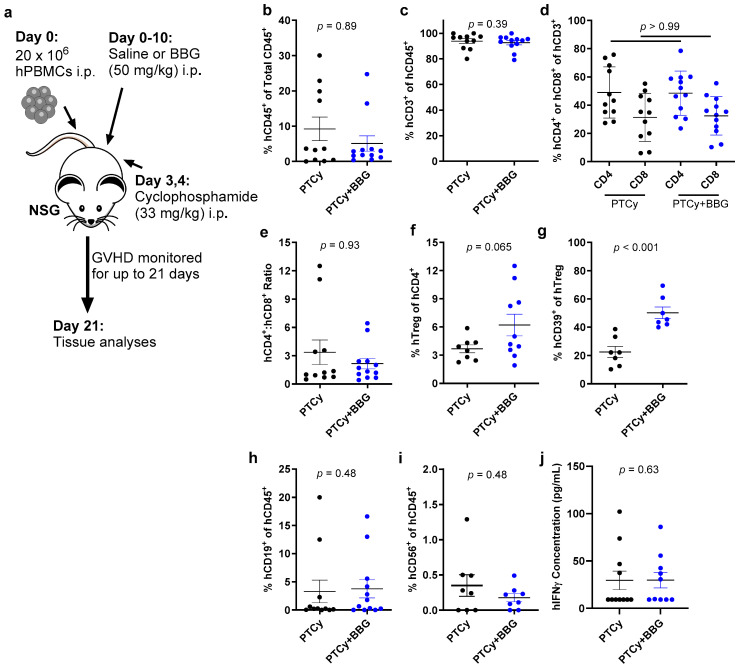
PTCy with BBG increases human CD39^+^ Tregs compared to PTCy alone on Day 21. (**a**) Schematic representation of the humanised mouse model of GVHD. (**b**–**j**) Spleen suspensions from humanised mice on Day 21 were examined by flow cytometry to determine the proportions of (**b**) human (h) CD45^+^ leukocytes, (**c**) hCD3^+^ T cells, (**d**) hCD4^+^ or hCD8^+^ T cells, (**f**) hCD4^+^hCD25^+^hCD127^lo^ Tregs, (**g**) hCD39^+^ Tregs, (**h**) hCD19^+^ B cells or (**i**) hCD56^+^hCD3^−^hCD19^−^ NK cells. (**e**) The ratio of hCD4^+^ to hCD8^+^ T cells was calculated from (**d**). (**j**) Serum hIFNγ concentrations were examined by a flow cytometric LEGENDPlex kit. (**b**–**h**,**j**) Data are presented as mean ± SEM. Symbols represent individual mice (*n* = 7–11 PTCy + BBG; *n* = 7–11 PTCy).

**Figure 5 ijms-25-01775-f005:**
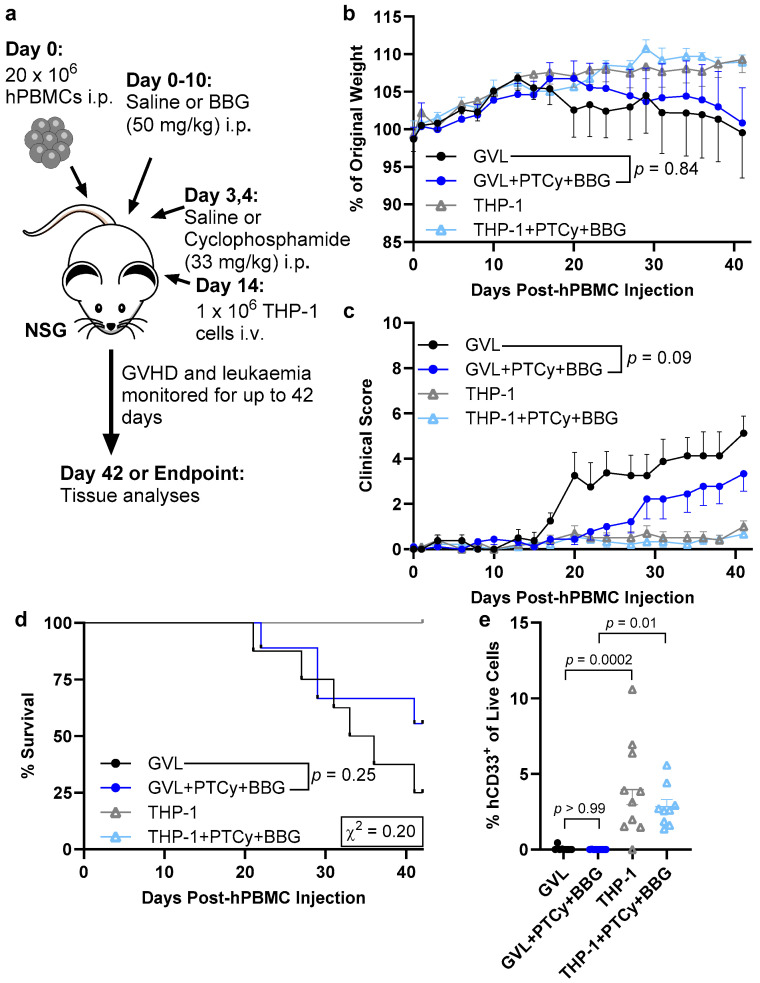
PTCy with BBG does not compromise GVL immunity. (**a**) Schematic representation of the humanised mouse model of GVHD/GVL. (**a**–**e**) NSG mice were humanised with 20 × 10^6^ human (h) PBMCs (*n* = 4 donors; presented as GVL groups) (or with PBS) on Day 0, treated with cyclophosphamide (33 mg/kg) or PBS (Days 3 and 4), and BBG (50 mg/kg) or saline (Days 0–10) then injected with 1 × 10^6^ THP-1 leukaemia cells (Day 14). Mice were assessed thrice weekly for (**b**) weight change, (**c**) clinical score, and (**d**) survival. (**e**) The proportions of hCD33^+^ THP-1 AML cells in liver suspensions were determined by flow cytometry. (**b**,**c**,**e**) Data are presented as mean ± SEM. (**d**) Data are presented as percent survival, χ^2^ test for mortality. (*n* = 8, GVL; *n* = 9, GVL + PTCy + BBG; *n* = 10, THP-1; *n* = 9, THP-1 + PTCy + BBG).

**Figure 6 ijms-25-01775-f006:**
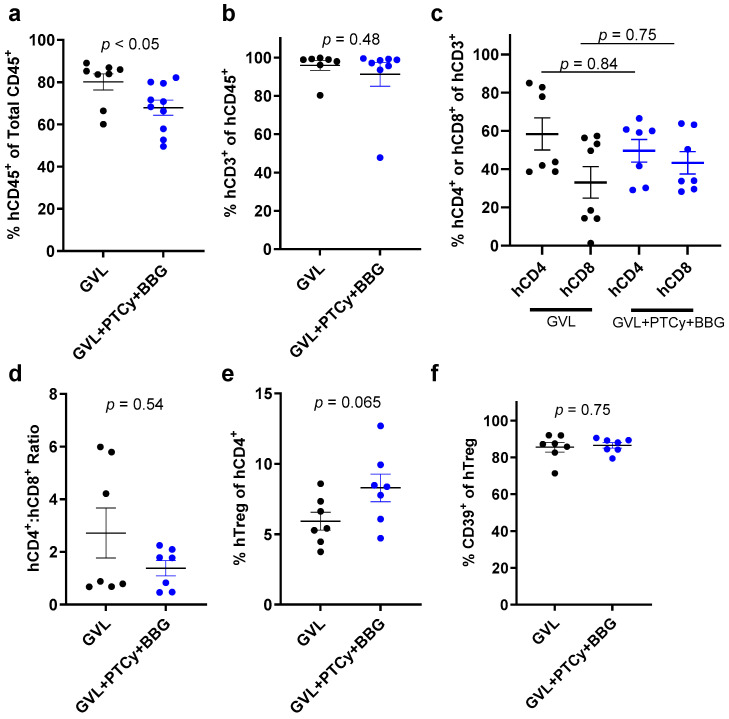
PTCy with BBG reduces proportions of splenic human leukocytes but not immune cell subsets compared to PTCy in GVL immunity at endpoint. (**a**–**f**) Spleen suspensions from treated mice at endpoint (Figure 5) were examined by flow cytometry to determine the proportions of (**a**) human (h) CD45^+^ leukocytes, (**b**) hCD3^+^ T cells, (**c**) hCD4^+^ or hCD8^+^ T cells, (**e**) hCD4^+^hCD25^+^hCD127^lo^ regulatory T cell (hTreg), or (**f**) hCD39^+^ Tregs. (**d**) The ratio of hCD4^+^ to hCD8^+^ T cells was calculated from (**c**). (**a**–**f**) Data are presented as mean ± SEM. Symbols represent individual mice (*n* = 8, GVL; *n* = 9, GVL + PTCy + BBG).

**Figure 7 ijms-25-01775-f007:**
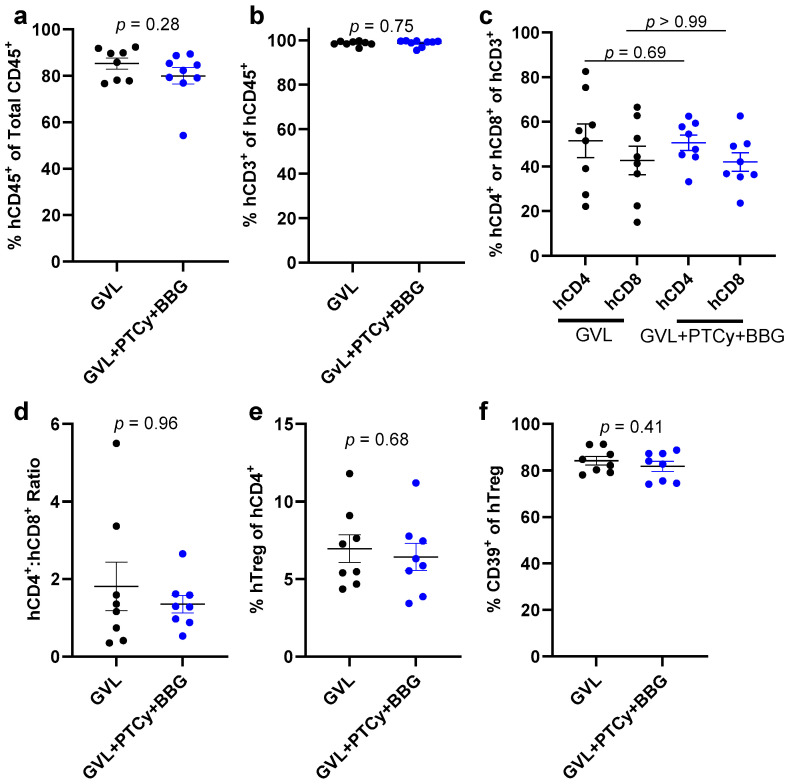
PTCy with BBG does not alter proportions of liver human leukocytes or immune cell subsets compared to PTCy in GVL immunity at endpoint. (**a**–**f**) Liver suspensions from treated mice at endpoint (Figure 5) were examined by flow cytometry to determine the proportions of (**a**) human (h) CD45^+^ leukocytes, (**b**) hCD3^+^ T cells, (**c**) hCD4^+^ or hCD8^+^ T cells, (**e**) hCD4^+^hCD25^+^hCD127^lo^ regulatory T cell (hTreg), or (**f**) hCD39^+^ Tregs. (**d**) The ratio of hCD4^+^ to hCD8^+^ T cells was calculated from (**c**). (**a**–**f**) Data are presented as mean ± SEM. Symbols represent individual mice (*n* = 8, GVL; *n* = 9, GVL + PTCy + BBG).

**Table 1 ijms-25-01775-t001:** GVHD/leukaemia scoring system.

Score ^1^	0	1	2	3
Acute weight loss ^2^	<5%	5–9.9%	10–14.9%	>15%
Chronic weight loss ^3^	<5%	5–6.9%	7–14.9%	>15%
Posture	Normal	Slight hunching	Moderate hunching	Severe hunching
Activity	Normal	Slightly reduced activity	Reduced activity and slow movement	Slow to no movement with reduced gait or shuffling
Hindlimb function	Normal gait	Unable to stand on hindlimbs	Mild paresis	Severe paresis
Fur	Normal	Mild to moderate ruffling	Severe ruffling and hair loss	>50% hair loss
Skin integrity	Normal	Scaling of paws and/or tail	Scaling of additional areas	Denuded skin

^1^ Mice were monitored daily if they obtained a score of 2 in any category (except fur) and euthanised if they obtained a score of 3 in any category. ^2^ Acute weight loss was calculated from the greatest weight over the past 7 days. ^3^ Chronic weight loss was calculated from Day 0 weight.

## Data Availability

The data supporting reported results are available from the corresponding author (R.S.) upon reasonable request.

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
