# Peer review of "Post-Transplant Cyclophosphamide Combined with Brilliant Blue G Reduces Graft-versus-Host Disease without Compromising Graft-versus-Leukaemia Immunity in Humanised Mice"

_ijms, 2024, doi:10.3390/ijms25031775_

Round 1
Reviewer 1 Report
Comments and Suggestions for Authors
In this manuscript, the authors investigated the combination of PTCy and P2X7 antagonism (BBG) in the control of GVHD in humanized mice. They observed that though the clinical GVHD development was comparable between PTCy with BBG and PTCy alone groups, the combined treatment reduced liver GVHD. They also found an increase in human splenic CD39+ Tregs and a decrease in human serum interferon-γ concentrations in PTCy with BBG mice. In addition, the combined treatment did not compromise the GVL in the mice liver. The data is interesting; however, the manuscript needs improvement before publication. Below are some comments/concerns.
1. P2X7 plays a critical role in dendritic cell functions; however, the humanized mice using NSG mice injected with PBMCs mainly maintained T cells (even 90% after several weeks), but minimal DCs. DCs are important in GVHD (such as some of pDCs) and GVL. NSG-SGM mice strain might be a better choice.
2. The authors observed that PTCy with BBG reduced liver GVHD, what is the level of cytokines in the liver? (only serum systemic expression was provided).
3. In figure 4, only the frequency of CD39+ Treg in the spleen was examined; what is the frequency of CD39+ Treg in the liver?
4. Figure 5, though GVHD was observed only in the liver, the GVL (% of CD33+ THP-1) should also be evaluated in other organs, such as blood, bone marrow, which are the initial and essential anatomical locations for acute myeloid leukemia.
Reviewer 2 Report
Comments and Suggestions for Authors
In their manuscript, the authors show that PTCy combined with BBG treatment reduces liver GVHD compared to PTCy alone without compromising GVL immunity to human THP-1 AML cells. They also show that BBG increases human CD39+ Tregs and decreases IFNg serum concentration. The measurements are carefully described, carried out and illustrated. Although the mechanisms of the observed effects are still unknown, the reported findings represent a logical addition to similar previous examinations and can potentially provide important indications for GVH therapies.
The only problem I have with the conclusion of the authors that P2X7 antagonism is responsible for their findings. This statement is based only on the use of the unspecific antagonist BBG, which also blocks P2X2 and P2X4 receptors, known to be expressed in cells of the immune system (Khakh et al., Pharmacol. Rev. 53). Therefore, either the authors show that the same effects of BBG can be obtained by a more specific P2X7 antagonist or the phrase "P2X7 receptor antagonism" is replaced with "BBG" in the title.
Minor point:
P12 paragraph 4: “In conclusion, PTCy combined with P2X7 antagonism … without promoting leukemia relapse.”
This paragraph should be omitted, since the same statement is already found in the Abstract.
Author Response
Thank you for the review and positive response to our original research manuscript.
The only problem I have with the conclusion of the authors that P2X7 antagonism is responsible for their findings. This statement is based only on the use of the unspecific antagonist BBG, which also blocks P2X2 and P2X4 receptors, known to be expressed in cells of the immune system (Khakh et al., Pharmacol. Rev. 53). Therefore, either the authors show that the same effects of BBG can be obtained by a more specific P2X7 antagonist or the phrase "P2X7 receptor antagonism" is replaced with "BBG" in the title.
Reply: We acknowledge that BBG is not specific for P2X7, so we have changed the title as requested. Further to this and since P2X7 is no longer in the title, we swapped “Brilliant Blue G” for “P2X7” as a keyword.
Minor point: P12 paragraph 4: “In conclusion, PTCy combined with P2X7 antagonism … without promoting leukemia relapse.” This paragraph should be omitted, since the same statement is already found in the Abstract.
Reply: Given International Journal of Molecular Biosciences encourages a conclusion and this is a minor point we have opted to retain this paragraph but have replaced “P2X7 antagonism” with “BBG” given the point above.
Kind regards,
Ronald Sluyter

Round 2
Reviewer 1 Report
Comments and Suggestions for Authors
The questions raised by this reviewer were addressed sufficiently. Thank you.